# Active control of anapole states by structuring the phase-change alloy $Ge_2Sb_2Te_5$

Jingyi Tian[1,2], Hao Luo[1], Yuanqing Yang [3], Fei Ding [3], Yurui Qu[1,4], Ding Zhao[5], Min Qiu [1,6,7] & Sergey I. Bozhevolnyi [3]

High-index dielectric nanoparticles supporting a distinct series of Mie resonances have enabled a new class of optical antennas with unprecedented functionalities. The great wealth of multipolar responses has not only brought in new physical insight but also spurred practical applications. However, how to make such a colorful resonance palette actively tunable is still elusive. Here, we demonstrate that the structured phase-change alloy $Ge_2Sb_2Te_5$ (GST) can support a diverse set of multipolar Mie resonances with active tunability. By harnessing the dramatic optical contrast of GST, we realize broadband ($\Delta\lambda/\lambda \sim$ 15%) mode shifting between an electric dipole resonance and an anapole state. Active control of higher-order anapoles and multimodal tuning are also investigated, which make the structured GST serve as a multispectral optical switch with high extinction contrasts (>6 dB). With all these findings, our study provides a new direction for realizing active nanophotonic devices.

[1] State Key Laboratory of Modern Optical Instrumentation, College of Optical Science and Engineering, Zhejiang University, Hangzhou 310027, China.
[2] Department of Applied Physics, Royal Institute of Technology, KTH, 10691 Stockholm, Sweden. [3] SDU Nano Optics, University of Southern Denmark, Campusvej 55, DK-5230 Odense, Denmark. [4] Department of Physics, Massachusetts Institute of Technology, Cambridge, MA 02139, USA. [5] DTU Danchip/ Cen, Technical University of Denmark, Kongens Lyngby 2800, Denmark. [6] School of Engineering, Westlake University, 18 Shilongshan Road, Hangzhou 310024, China. [7] Institute of Advanced Technology, Westlake Institute for Advanced Study, 18 Shilongshan Road, Hangzhou 310024, China. These authors contributed equally: Jingyi Tian, Hao Luo. Correspondence and requests for materials should be addressed to Y.Y. (email: yy@mci.sdu.dk) or to M.Q. (email: minqiu@zju.edu.cn)

Ever since the seminal work of Mie[1], light scattering by resonant small particles has attracted a vast amount of attention in many branches of physics[2]. The intention to control and manipulate light by fully exploiting the advantages from scattering resonances, particularly at the nanometer scale, has stimulated the emergence of modern nanophotonics[3] and spawned a myriad of applications ranging from biochemistry to information technology[4]. In this context, low-loss, high-index dielectric or semiconductor nanostructures featuring a diverse set of optical resonances are currently in the spotlight of research as they can serve as versatile and CMOS-compatible building blocks for various photonic devices[5–9]. Besides the practical advances, studies on dielectric nanoresonators have also brought new insight into fundamental physics. Recent experimental investigations on the scattering response of Si nanoparticles have not only shown conventional radiant modes such as magnetic dipole (MD) or electric dipole (ED) resonances[10–12], but also revealed the underlying physics of an intriguing scattering "dark state", i.e., anapole state, characterized by a pronounced minimum in the scattering spectra and an associated maximum in the near-field energy[13–15]. Such a suppressed scattering state stems from two antiphased electric and toroidal dipole moments, whose radiation patterns are identical to each other and thereby interfere destructively in the far field. This unique behavior of anapole states shortly unveil its tantalizing potential in many scenarios such as cloaking[16,17], nanoscale lasers[18], field enhancements[19,20], energy guiding[21], harmonic generations[22–24] and metamaterials[25–27].

However, despite the great wealth of optical resonances and rendered interesting phenomena, how to actively tune these responses and further switch among them remains a daunting challenge. This is because the induced near fields of dielectric nanoparticles, unlike their plasmonic counterparts, are mainly inside the structures and thereby only mildly sensitive to the change of external environments. For the same reason, the majority of research to date still focuses on passive structures, whose functionalities are set in during fabrication and cannot be altered afterward. Whereas there is a growing recognition of the need to realize active dielectric components, most of the published reports so far only display modest resonance shifts[28–35]. For instance, a pioneering work using liquid crystals as embedding media[29] generates a maximum spectral shift $\Delta\lambda \approx 40$ nm at resonance wavelength $\lambda \approx 1550$ nm, corresponding to a relative resonance tuning $\Delta\lambda/\lambda$ merely 2.6%. A very recent study utilizing the thermo-optic effect of Si achieves a resonance shift $\Delta\lambda \approx 30$ nm at wavelength $\lambda \approx 1500$ nm under an external temperature around 300 °C[33]. Indeed, given the multitude of resonances and follow-up functionalities offered by high-index dielectrics, tuning one or a few spectral peaks with limited ranges does not sufficiently employ all the benefits from such a fruitful playground. While attempts are also being made to obtain wideband tunability spanning over one linewidth[36–38], active mode switching between or among different dielectric resonances is still an unexplored conundrum.

In this work, we realize broadband and controllable mode shifting between distinct scattering states by structuring the phase change alloy $Ge_2Sb_2Te_5$ (GST-225, simply GST hereafter). Owing to its striking electrical and optical contrasts between amorphous and crystalline phases, GST has been widely used in commercial memory applications and was recently introduced into the nanophotonics community[39]. In contrast to the aforementioned tuning mechanisms such as using liquid crystals or temperature tuning, GST affords a different, non-volatile approach where the induced optical change remains stable even after the removal of external stimuli. So far, most of the research employs GST in the form of thin films functioning as surrounding media for metallic structures[40–49]. For instance, Chen et al. realized stepwise tuning of the lattice resonance in a hybrid plasmonic system consisting of a gold disk array and an underneath GST thin film[40]. The characteristics of such configuration thus are still dominated by lossy plasmonic resonances. Although there are also exciting developments using GST itself as integrated optical constituents[50–55], detailed investigations on the fundamental optical properties of GST nanostructures are surprisingly lacking[39,56]. In fact, among all the phase-change chalcogenides, GST features one of the highest refractive indices in its amorphous states[57], which satisfies the essential prerequisite for constructing dielectric nanoantennas with strong Mie resonances. Here, we perform a thorough multipole analysis to examine the optical response of standalone GST nanostructures both theoretically and experimentally. For the first time, we demonstrate that the high refractive index and the low loss of GST empower its nanostructures to support MD, ED and anapole states, in a similar manner as other enticing dielectrics such as Si and Ge. Meanwhile, the distinctive tunability of GST makes all these resonances actively controllable. By exploiting the intermediate phases of GST, we show progressive mode shifting between scattering bright and dark states over an extremely broadband region ($\Delta\lambda/\lambda \sim 15\%$). Multimodal shifting among higher-order ED and anapole states is also manifested, naturally making the investigated GST structures function as a multispectral optical switch with high extinction contrasts (>6 dB) as well as multi-level control abilities. Hence, by discovering the concealed portfolio of actively controllable resonances in GST nanostructures, our findings establish a new basis for designing active optical components and pave the way towards metadevices with tunability on demand[58].

## Results

**Mie resonances in GST nanospheres**. To analyze the electromagnetic response of GST nanostructures, we start our investigation by considering the most general case for analytical treatments: a GST sphere situated in the vacuum (see Methods). Here we focus on the mid-infrared range given the high refractive index, substantial optical contrast, and relatively low loss of the GST material[57]. Fig. 1a conceptually illustrates two representative scattering states, i.e., ED and anapole states, supported by the GST sphere. At the ED resonance, the induced ED moment **p** generates a considerable dipolar radiation, giving rise to a scattering bright state. By contrast, the anapole mode features an induced poloidal current of electric fields inside the particle, associated with a torus of circulating magnetic fields[17]. Such an intricate field distribution is characterized by a significant suppression of far-field scattering. Therefore, this phenomenon is also commonly referred to as a scattering dark state. Based on the Mie theory, the spectral positions of these two modes are heavily dependent on the refractive index of the particle. Therefore, shifting between the modes could be realized by introducing a delicate amount of index change to the GST sphere. To this end, thermal, electrical, and optical stimuli could be utilized to induce controllable phase transformations in the GST material, making it possible to achieve not only the transition between amorphous and crystalline states but also among intermediate (semicrystalline) levels[39]. The scattering response of a GST sphere with a fixed radius $R = 450$ nm and varying crystallinities $C$ is depicted in Fig. 1b. With a stepped phase change $\Delta C = 25\%$, a progressive shifting between scattering maxima and minima can be readily observed. A detailed multipole analysis further shows that the scattering maxima and minima are ambiguously attributed to the ED and anapole states, respectively (Fig. 1c). In particular, the anapole state (denoted as A) corresponds to the minimum partial scattering contributed by the spherical ED, as unraveled by ref. [13].

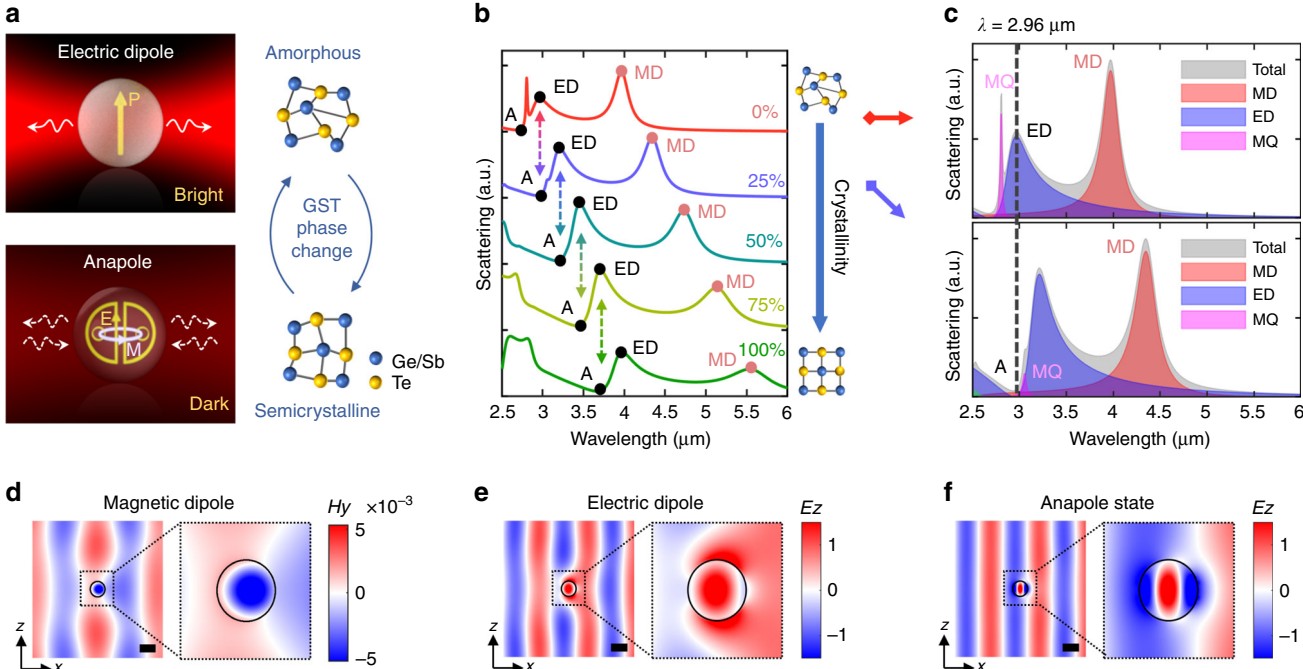

**Fig. 1** Shifting between scattering bright and dark states in a GST sphere. **a** Conceptual illustration of an electric dipole resonance and an anapole excitation in a GST sphere. Shifting between these two states can be realized by introducing an intermediate phase transition in the GST material. **b** Normalized scattering spectra of a GST sphere ($R = 450$ nm) with different crystallinities. A continuous shifting between ED and anapole states (denoted as A) can be clearly observed. **c** Analytical multipole analysis of two representative scattering spectra of amorphous GST (aGST) and 25%-crystalline GST (25%-cGST) spheres. **d**, **e** Magnetic and electric field distributions of the aGST sphere excited at the two scattering bright states, i.e., MD ($\lambda = 3.95$ μm) and ED ($\lambda = 2.96$ μm) resonances. **f** Electric field distributions of the 25%-cGST sphere at the anapole excitation ($\lambda = 2.96$ μm). The scale bars in **d**–**f** represent 1 μm

Other multipolar resonances, such as MD and magnetic quadrupole (MQ) are also distinctly manifested. The prolific multipolar effects of the GST sphere come from its notably high index ($n_{aGST} > 4$, $n_{cGST} > 6$, see Supplementary Note 1 and Supplementary Fig. 1), which is of central importance to the field of all-dielectric nanophotonics. Besides the two representative examples ($C = 0\%$ and $C = 25\%$) shown in Fig. 1c, in Supplementary Fig. 2a we provide a detailed multipole analysis for other crystalline phases of GST, clarifying that the progressive shifts of the multipolar effects indeed take place in the GST sphere with all different crystallinities. The effects of the crystallinity and material loss on the bandwidth of different Mie states are also discussed in Supplementary Note 2 and shown in Supplementary Fig. 3.

In Fig. 1d–f we plot the field distributions of three typical scattering states excited in the GST sphere. The incident planewave propagates along the $x$-axis with the electric fields polarized along the $z$–direction. At MD resonance (Fig. 1d), the magnetic fields are concentrated at the center of the sphere with noticeable scattered fields while the associated electric fields are circulating around the center (Supplementary Fig. 2b). At ED resonance (Fig. 1e), the electric fields exhibit a dipolar response parallel to the incident polarization ($z$-axis). An appreciable scattering process also occurs, as shown in Fig. 1e. However, in contrast to the two bright modes, the GST sphere seems to be transparent with imperceptible scattering at the anapole state (Fig. 1f). The displayed field distribution with antiphased $E_z$ (Fig. 1f) and two field zeros (Supplementary Fig. 2c) along the $x$–direction is indeed the signature of an anapole excitation, as discussed in our previous work[19,20]. Hence, the rich collection of active Mie resonances supported by GST spheres is revealed. We note that, given the plethora of phenomena caused by Mie resonances and associated multipolar effects[5–9], active tunability can straightforwardly be implemented into many existing

applications by utilizing GST resonators. For instance, in Supplementary Note 3 and Supplementary Fig. 4, we explore the possibilities to realize the mode shifting between the MD and ED resonances. Such a tunability also allows us to actively control the directionality of corresponding far-field scattering. Among all the promising opportunities, here we focus our attention on the mode shifting between ED and anapole states.

**Broadband mode shifting between scattering bright and dark states**. Next, we examine the spectral response and the tuning range of the mode shifting effect. Since the ED resonance and the anapole state are related to the maximum and the minimum of the ED contribution, they are only related to the multipole coefficient $a_1$ which is the function of the radius $R$, the crystallinity $C$ and the incident wavelength $\lambda$ (see Methods for more details). Therefore, the spectral positions of the two modes are determined by both the crystallinity $C$ and the geometric size $R$. In this regard, we first consider a GST sphere with an invariant radius ($R = 450$ nm) and continuously change its crystallinity $C$. The two-dimensional map of its scattering efficiency $Q_{scat}$ is plotted in Fig. 2a. The pronounced scattering maxima and minima (marked by the white dashed lines for eye guidance) undergo continuous redshifts with the increasing crystallinity $C$. To precisely identify the position of ED and anapole states, we further analytically obtain the conditions for the two modes by solving the following sets of equations and inequalities: (1) $|a_1(\lambda, R, C)|' = 0$, $|a_1(\lambda, R, C)|'' < 0$ for the ED resonance, and (2) $|a_1(\lambda, R, C)|' = 0$, $|a_1(\lambda, R, C)|'' > 0$ for the anapole state. With a fixed $R = 450$ nm, the solution set ($\lambda$, $C$) is shown in Fig. 2b. The shaded area between the ED (red) and the anapole (blue) lines represents the effective region where the GST sphere can support the mode shifting between a ED and an anapole state. An ultra-broadband response of the shifting effect can be observed as the

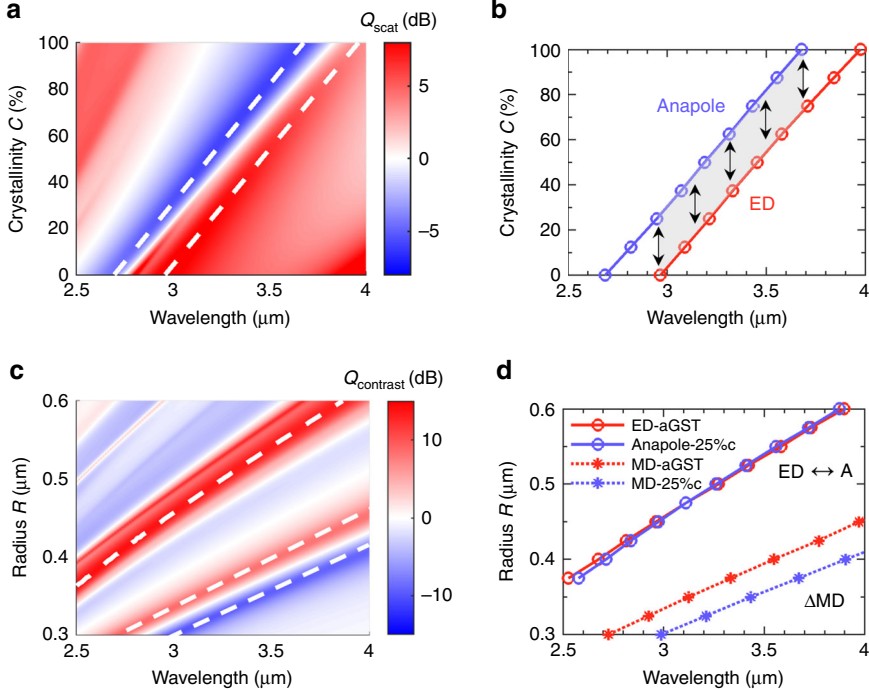

**Fig. 2** Broadband tuning and shifting of the multipolar scattering states in GST spheres. **a** Scattering efficiency $Q_{scat}$ of a single GST sphere with a fixed radius $R = 450$ nm and varying crystallinities $C$. Two white dashed lines indicate the scattering maxima and minima corresponding to the ED and anapole states. **b** Analytically solved ED and anapole conditions. Shifting between these two states can be realized within an ultrabroad bandwidth over 700 nm (the gray shaded area) by exploiting the intermediate levels of GST spheres. **c** Scattering cross-sectional contrast between aGST and 25%-cGST spheres with varying radii $R$. Three white dashed lines show the most noticeable scattering contrasts. **d** Analytically solved MD, ED, and anapole conditions to account for the scattering contrast between the aGST and 25%c-GST spheres. The ED and anapole positions are almost perfectly overlapped in a wavelength range spanning around 1.5 μm

area spans over $\Delta\lambda > 700$ nm along the x-axis, corresponding to a fractional bandwidth $\Delta\lambda/\lambda > 20\%$. Similarly, the height of the area along the y-axis indicates the amount of a phase change needed to implement the mode shifting. Interestingly, such an amount is nearly constant over the whole spectral range, meaning that the presented mode shifting functionality (from the ED to the ana-pole state) can be attained by simply introducing a fixed phase change ($\Delta C \approx 25\%$) to the GST nanosphere with an arbitrary crystallinity $C$ below 75%.

Then we study the impacts of the geometric size on the mode shifting effect. To this end, we consider the spheres with different radii $R$ and investigate the scattering contrast of these spheres at two distinct crystalline phases: amorphous ($C = 0\%$) and an intermediate phase ($C = 25\%$). The scattering contrast $Q_{constrast}$ is defined as the ratio between the scattering efficiencies of the spheres at the two phases. The 2D scattering maps of each phase are provided in Supplementary Note 4 and Supplementary Fig. 5. Noticeable spectral shifts of all the multipolar responses can be clearly seen when the GST spheres experience a phase transformation from the amorphous (Supplementary Fig. 5a) to the semi-crystalline phase (Supplementary Fig. 5b), or vice versa. Consequently, three substantial scattering contrasts over 10 dB can be found on the map of $Q_{constrast}$ (marked by the white dashed lines in Fig. 2c). A multipole analysis (Fig. 2d) further explicitly shows that these dramatic scattering contrasts are mainly attributed to two mechanisms: (1) the spectral shifts of the MD resonances, giving rise to transitions between resonant and non-resonant states; (2) the mode "switching" between the ED and the anapole states, giving rise to the transition from scattering maxima to minima. In particular, we find that the spectral positions of the ED and the anapole states are almost perfectly overlapped with each other in the entire wavelength range of

interest ($\Delta\lambda \approx 1500$ nm). Therefore, given a cluster of GST spheres with various radii, by introducing a fixed amount of phase change (here $\Delta C = 25\%$), all the structures possessing different ED resonance wavelengths would exhibit the same functionality shifting from the ED to the corresponding anapole state. Thus, the mode shifting effects in GST nanostructures can be maintained in a broadband range regardless of the original resonance wavelengths. Such a nearly "non-dispersive" behavior may find its applications in many interesting aspects. For instance, a major challenge nowadays to realizing actively tunable metasurfaces lies in the fact that metasurfaces are usually composed of plasmonic or dispersive meta-atoms with different sizes and different resonant wavelengths. Therefore, a uniform optical change across the interface does not guarantee that the metasurface can sustain its important functions (e.g., focusing, invisibility, polarization conversion, etc) after active tuning. By contrast, GST resonators with nearly non-dispersive behaviors may provide a promising solution to overcome this issue.

**Experimental realization of actively tunable scattering states.** To verify the proposed concepts, we then perform experiments with GST nanostructures. Given the ease of fabrication and convenience for observing anapole states, truncated GST disks were fabricated by using E-beam lithography, magnet sputtering deposition, and standard lift-off process (see Methods). The geometric profile of the fabricated sample was measured by atomic force microscopy (see Supplementary Note 5, Supplementary Figs. 6 and 7). The height of the disks is 220 nm and the ratio between the bottom and top radii of the GST disks is set to 2 μm for all the samples based on preliminary numerical designs. Thus, in the following, we can simply use the bottom radius $R$ to

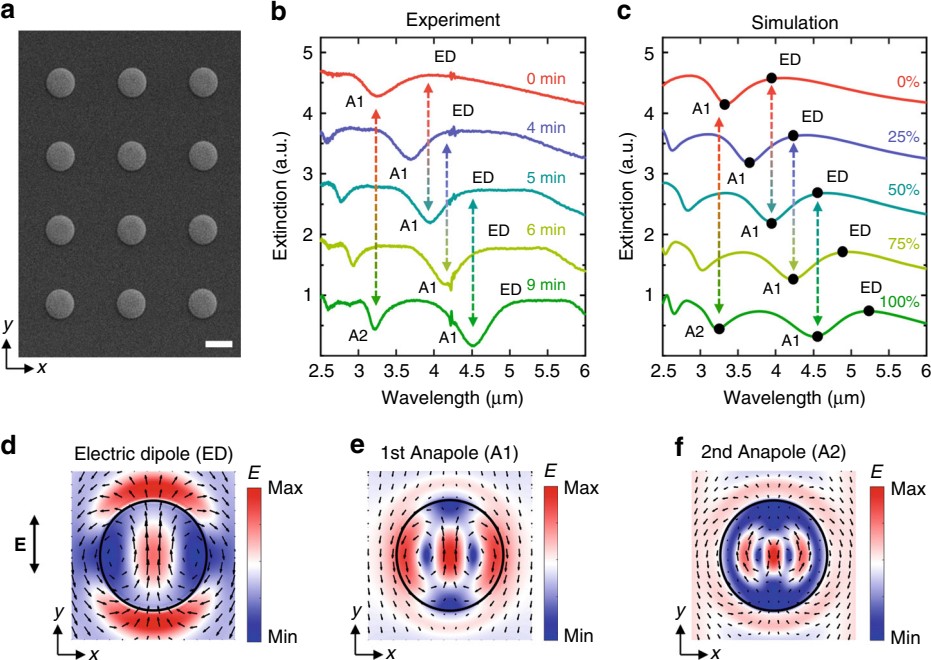

**Fig. 3** Experimental realization of active tunable anapole states in structured GST nanodisks. **a** SEM image of the fabricated GST disks. The scale bar represents 2 μm. **b**, **c** Experimental (**b**) and simulated (**c**) extinction spectra of GST disks with radius $R = 1$ μm and height $H = 220$ nm. A1 and A2 denotes the 1st-order and the 2nd-order anapole states, respectively. **d–f** Near-field distributions of the ED, A1, and A2 states in the middle planes of the aGST, 50%–cGST and 100%–cGST disks, respectively

describe the geometric feature of the disks. An SEM image of fabricated GST nanodisk array with $R = 1$ μm is shown in Fig. 3a. A pitch $g$ of 3 μm was chosen to avoid coupling between adjacent nanostructures. The influences of the pitch size, the absorption loss of the GST material, and the substrate $CaF_2$ are thoroughly examined and can be referred to Supplementary Note 6 and Supplementary Figs 8 and 9.

To introduce different amounts of phase change and realize intermediate phases of GST, here we applied thermal stimuli by heating the sample on a hotplate at a fixed temperature 145 °C but with different amounts of time. The extinction spectra of the GST disks after different annealing time are presented in Fig. 3b. We can clearly see that the GST disks indeed support notable extinction maxima and minima with evident shifting between these peaks and valleys, in a similar manner to the GST spheres. Corresponding numerical results are provided in Fig. 3c and a good agreement between the experimental and simulation results can be observed. Here we estimate the crystallinity $C$ by matching the spectral position of the anapole states in the experimental, similar to ref. [40]. Slight deviations in relative resonance strengths and linewidths might be attributed to the non-zero (although small) oblique incident angle provided by the applied objective, lattice effect, or a slight overestimation of the material loss used in the simulation. Multipole decomposition of the spectral response (see Methods) further clearly points out that the extinction maxima and minima are exactly correlated with the ED and anapole states, respectively. In particular, we find that, once a phase change of $\Delta C = 50\%$ is introduced (e.g., from $C = 0\%$ to $C = 50\%$, $C = 25\%$ to $C = 75\%$, or $C = 50\%$ to $C = 100\%$), the GST disks would always undergo a shift between the ED and the first-order anapole state (A1). Such a mode shifting effect thus can be achieved in a broadband region over 600 nm (from 3.9 μm to 4.6 μm, corresponding to $\Delta\lambda/\lambda \sim 15\%$), which is remarkably consistent with our previous theoretical investigations on the GST spheres.

Besides the existence of the ED and A1 states, the large diameter-to-height ratio of the disks also enables the emergence of higher-order ED and anapole states, such as the second-order anapole state (A2) supported by the 100%–cGST disk. The near-field distributions of the ED, A1, and A2 states are depicted in Fig. 3d–f. One can observe that the A2 state supports two pairs of poloidal currents which result in four field zeros along the $x$-axis, indicating a clear combination of the A1 state and an accompanied standing wave character. This phenomenon can be explained by the generation of hybrid Mie–Fabry–Perot modes[26] or the superposition of several internal modes[59]. Hence, the A2 state possesses a stronger field confinement within the disk volume compared to the A1 state, which leads to a higher concentration of internal energy[19]. Therefore, such higher-order anapoles could exhibit their unique advantages over their fundamental counterparts, particularly in scenarios such as harmonic generation[23] and field enhancement[20]. Interestingly, we find that the shifting between the A1 and the A2 states can be also realized by changing a GST disk from its amorphous phase to its crystalline phase (Fig. 3b, c).

**Multimodal and broadband tuning behavior.** After revealing the higher-order anapole states, we then thoroughly examine the multimodal response and associated tunable behavior of the GST disk. In Fig. 4a we plot the simulated 2D scattering map of the GST disk with different crystallinities $C$ varying from 0% to 100%. Scattering bright and dark states appear alternately across the spectra, indicating the existence of higher-order ED (denoted as ED2, ED3) and anapole states (see Supplementary Fig. 9 for detailed multipole decomposition). The experimentally measured spectral positions of these states coincide well with the simulation results, as shown in Fig. 4b. One can clearly find that, besides the demonstrated mode shifting response, various multimodal tuning and shifting can be realized among the presented scattering states,

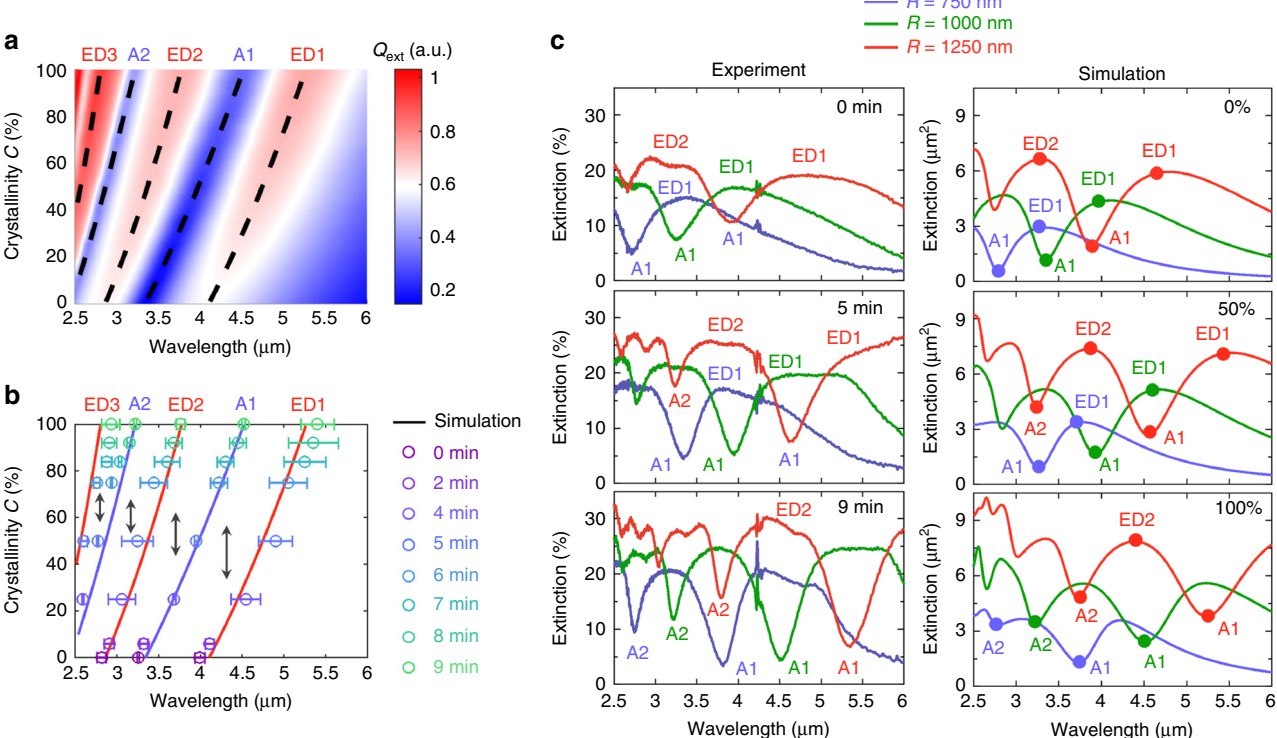

**Fig. 4** Multimodal and broadband tuning effects of GST disks with varying crystallinities and radius. **a** Simulated 2D extinction map of GST disks with a fixed radius $R = 1\,\mu m$ and varying crystallinities $C$. The dashed lines indicate the series of bright and dark states, i.e., ED1, A1, ED2, A2, ED3. **b** Numerically solved conditions (solid lines) and experimentally measured positions (circles) for different bright and dark states under different crystallinities. **c** Experimental (left) and simulated (right) extinction spectra of structured GST disks with different disk radius. Mode shifting effects such as from ED to A1 or from A1 to A2 are demonstrated in a broadband region over $1\,\mu m$

e.g., the possible mode shifting from A1 to ED2, from ED2 to A2, or from A2 to ED3 modes, etc. It is also worth mentioning that, mode shifting can not only occur between a bright and a dark state but also take place within two bright (e.g., ED2 and ED3) or two dark (e.g., A1 to A2) states. Compared to the GST sphere, the fabricated GST disk possesses much more fruitful tuning and mode shifting phenomena due to its additional broken symmetry. Therefore one may naturally expect to unlock numerous new possibilities by structuring GST into different resonant shapes.

Next, we investigate GST disks with different radii $R$. Experimental and simulation extinction spectra of the disks at three representative crystalline phases ($C = 0\%$, $C = 50\%$, and $C = 100\%$) are plotted in Fig. 4c. A good accordance between the experimental and simulation results can be seen for all the disks. In particular, when a phase change $\Delta C = 50\%$ is introduced, all the disks exhibit the same mode shifting response from the ED1 to the A1 states, despite their different ED1 resonance wavelengths $\lambda_{ED1}$ ranging from 3.2 $\mu m$ to 4.6 $\mu m$. Similarly, when a phase change $\Delta C = 100\%$ is introduced (from the amorphous phase to the crystalline phase), all the disks with different radii shift their A1 states to corresponding A2 states at different wavelengths $\lambda_{A1}$ spanning from 2.7 $\mu m$ to 3.7 $\mu m$. Hence, for both aforementioned mode shifting effects (ED1 to A1 and A1 to A2), nearly non-dispersive responses over 1 $\mu m$ are demonstrated. Once again, these results substantiate our previous theoretical investigations on GST spheres. It is also worth noting that other multimodal responses such as the shifting from A1 to ED2 and from ED2 to A2 are also sustained for all the disk sizes in a broadband region.

Finally, it is interesting to point out that the presented tunable scattering bright and dark states of the GST structures naturally lend themselves as an optical switch. In contrast to conventional

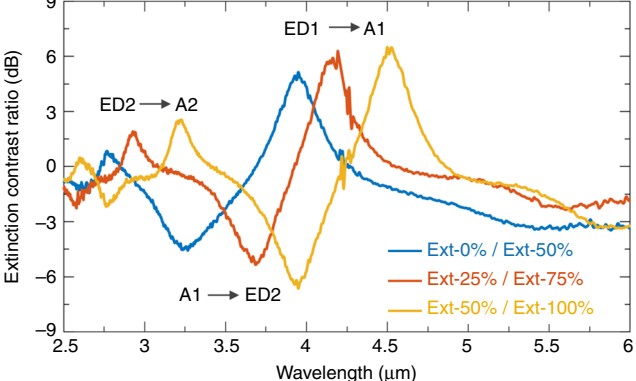

**Fig. 5** Multispectral optical switch with multi-level control capabilities. Three representative extinction contrasts of GST disks ($R = 1\,\mu m$, the same as in Fig. 3) with different crystallinities are plotted, which are between (i) $C = 0\%$ and $C = 50\%$ (blue), (ii) $C = 25\%$ and $C = 75\%$ (red), and (iii) $C = 50\%$ and $C = 100\%$ (yellow), respectively

optical switches which usually work at single wavelengths[41], here the multimodal tuning and shifting behavior of the GST disks endow themselves with a multispectral capability. As we can see from Fig. 5, once a 50% phase change is introduced (e.g., from $C = 0\%$ to $C = 50\%$, the blue curve), the GST disks will undergo three disparate switching at three different wavelengths simultaneously. Both "on" and "off" states of the optical switch thus could be achieved at the same time in different spectral regions. A high contrast ratio of 6 dB is obtained without any further geometry optimization. Moreover, due to the broadband response of the switching functionalities, such a multispectral characteristic

is sustained for different crystalline phases of the GST material (e.g., the red and the yellow lines in Fig. 5), which offers the possibilities for the optical switch to monitor or be controlled by multi-level external stimuli. Besides, similar to the case of GST spheres, other mode shifting effects could also be realized by exploiting intermediate phases in the GST disk arrays with properly designed geometry, a new fascinating direction that promises many future developments. In Supplementary Note 7 (and Supplementary Figs. 10 and 11), we exemplify these possibilities by providing a numerical demonstration of a GST metasurface for tunable beam steering.

## Discussion

In summary, we investigated the fundamental optical response of individual GST nanostructures by virtue of a rigorous multipole analysis. We revealed that the high index and the dramatic optical contrast of the GST material empower its nanostructures to support a distinct series of Mie resonances with active tunability. Since the entire field of all-dielectric meta-optics rests on multi-polar Mie resonances and their interference, by offering such a dynamic resonance palette, GST nanostructures thus can serve as a powerful platform for tunable nanodevices towards various applications including metasurfaces, biosensing, and nonlinear optics. In particular, we then demonstrated the spectral shifting between ED and anapole modes, corresponding to scattering bright and dark states, respectively. A broadband response of the mode shifting effect was examined and we showed that the ED-to-anapole shifting can be achieved in a GST nanodisk by simply introducing a phase change $\Delta C = 50\%$ at any given wavelengths between 3.9 μm to 4.6 μm and with an arbitrary crystallinity of the GST disk. Furthermore, we demonstrated the nearly non-dispersive behavior of the modes shifting response over 1 μm, indicating that a uniform phase change can enable structures with different sizes to exhibit the same tuning functionality, regardless of their distinct resonance wavelengths. In addition, the existence of higher-order ED and anapole states was presented with a thorough study on the multimodal shifting among all these states. As a proof-of-principle application, we demonstrated a multi-spectral optical switch with multi-level control capabilities. We note that our present work provides a systematic yet prototypical demonstration of the active multipolar effects in GST nanostructures via thermal annealing. To realize reversible tuning with electric or optical stimuli, further considerations and adaptions of the layout need to be taken into account, for example, the inclusion of electrodes and protective layers[39]. Nevertheless, given the plethora of unexplored possibilities hindered in the structured GST material, we envision that our results open an entirely new direction in active meta-optics. As a final remark, we mention that there is a wide selection of phase-change chalcogenides featuring extraordinary optical contrasts and low loss[57], holding the promise for further developments and other opportunities.

## Methods

**Analytical calculations for GST spheres**. To obtain the electromagnetic response of the GST spheres, we applied Mie theory[2] which offers the exact solution to the scattering problem and allows writing the scattering efficiency $Q_{scat}$ in the following simple form:

$$Q_{scat} = \frac{2}{k^2 R^2} \sum_{\ell=1}^{\infty} (2\ell + 1) \left[ |a_\ell|^2 + |b_\ell|^2 \right], \tag{1}$$

where $Q_{scat}$ is defined as the ratio between the scattering cross section and the geometrical cross-section of a sphere, namely $\pi R^2$. $k = 2\pi/\lambda$ is the wave number related to the incident wavelength $\lambda$. The contributed multipole coefficients $a_\ell$

(electric) and $b_\ell$ (magnetic) can be read as:

$$a_\ell = \frac{[D_\ell(nKR)/n + \ell/kR]\psi_\ell(kR) - \psi_{\ell-1}(kR)}{[D_\ell(nKR)/n + \ell/kR]\xi_\ell(kR) - \xi_{\ell-1}(kR)}, \tag{2}$$

$$b_\ell = \frac{[nD_\ell(nKR) + \ell/kR]\psi_\ell(kR) - \psi_{\ell-1}(kR)}{[nD_\ell(nKR) + \ell/kR]\xi_\ell(kR) - \xi_{\ell-1}(kR)}, \tag{3}$$

where $n$ is the refractive index of GST. $D_\ell(nKR)$ is defined as $D_\ell(nKR) = \psi'_\ell(nkR)/\psi_\ell(nkR)$, with $\psi_\ell(kR)$ and $\xi_\ell(kR)$ the Riccati-Bessel functions of the first and second kind. The total scattering of the GST spheres was calculated by considering the multipole contributions up to the quardupole order.

In all calculations, we adopted experimentally measured optical constants of GST (Supplementary Fig. 1). For any intermediate phases with a crystallinity $C$ ($0 \le C \le 1$), the dielectric constant $\varepsilon_{GST}(\lambda, C)$ of GST can be estimated by using the effective medium theories (EMTs). Out of various EMTs, here we choose to use the Lorentz-Lorenz relation as it is so far one of the most widely used approaches in the simulation of hybrid and isolated GST nanostructures[40,56]. Comparing to other EMTs such as the Maxwell-Garnett approximation[60], the maximum difference in the real part of the refractive index $\Delta n_0$ between different EMTs was found to be smaller than 5%. Therefore, our estimation is well within the acceptance of EMT approximation. The Lorentz-Lorenz relation can be expressed as follows[61]:

$$\frac{\varepsilon_{GST}(\lambda, C) - 1}{\varepsilon_{GST}(\lambda, C) + 2} = C \times \frac{\varepsilon_{cGST}(\lambda) - 1}{\varepsilon_{cGST}(\lambda) + 2} + (1 - C) \times \frac{\varepsilon_{aGST}(\lambda) - 1}{\varepsilon_{aGST}(\lambda) + 2}, \tag{4}$$

where $\varepsilon_{aGST}$ and $\varepsilon_{cGST}$ are the permittivities of amorphous and crystalline GST, respectively. Therefore, by applying Eq. (1–4), we can clearly identify ED and anapole states by treating the partial scattering of the electric dipole: $Q_{scat}|_{a_1}(\lambda, R, C)$.

**Numerical simulations and multipole decomposition**. We performed three-dimensional FDTD simulations with a commercial software package (Lumerical). The optical constants of the GST disks with different crystallinities was determined by the experimental ellipsometric data (Supplementary Fig. 1) and the Eq. (4). The refractive index of the substrate CaF₂ was set to 1.4[62]. A normal-incident total-field/scattered-field planewave source was utilized to calculate the extinction, scattering, and absorption cross-section of the GST disks. A mesh size of 10 nm was set over the whole volume of the GST disks. Perfectly matched layers were set as the boundaries to enclose the simulation area. To carry out multipole decomposition of the simulated spectra, a three-dimensional frequency-domain field monitor was used to record the electric fields $\mathbf{E}(\mathbf{r})$ at every discretized points $\mathbf{r}$ (coordinate respective to the disk's center) inside the disks. By defining the polarization current $\mathbf{J}(\mathbf{r}) = -i\omega\varepsilon_0[\varepsilon_r(\mathbf{r}) - 1]\mathbf{E}(\mathbf{r})$, the electric $a(\ell, m)$ and magnetic $b(\ell, m)$ spherical multipole coefficients can be calculated via the following formulae[63]:

$$a(\ell, m) = \frac{(-i)^{\ell-1}k\eta}{2\pi E_0} \frac{\sqrt{(\ell-m)!}}{\sqrt{\ell(\ell+1)(\ell+m)!}} \int \exp(-im\phi) \Big\{ [\psi_\ell(kr) + \psi''_\ell(kr)] P_l^m(\cos\theta)\hat{\mathbf{r}} \cdot \mathbf{J}(\mathbf{r})$$
$$+ \frac{\psi'_\ell(kr)}{kr} \Big[ \frac{d}{d\theta} P_l^m(\cos\theta)\hat{\theta} \cdot \mathbf{J}(\mathbf{r}) - \frac{im}{\sin\theta} P_l^m(\cos\theta)\hat{\phi} \cdot \mathbf{J}(\mathbf{r}) \Big] \Big\} d^3\mathbf{r}, \tag{5}$$

$$b(\ell, m) = \frac{(-i)^{\ell+1}k^2\eta}{2\pi E_0} \frac{\sqrt{(\ell-m)!}}{\sqrt{\ell(\ell+1)(\ell+m)!}} \int \exp(-im\phi) j_l(kr) \Big[ \frac{im}{\sin\theta} P_l^m(\cos\theta)\hat{\theta} \cdot \mathbf{J}(\mathbf{r})$$
$$+ \frac{d}{d\theta} P_l^m(\cos\theta)\hat{\phi} \cdot \mathbf{J}(\mathbf{r}) \Big], \tag{6}$$

where $E_0$ is the electric field amplitude of the incident plane wave; $\eta$ is the impedance of free space; $j_l(kr)$ is the spherical Bessel function of the first kind and $P_l^m(\cos\theta)$ is the associated Legendre polynomials. Thus the total scattering cross section $C_{scat}$ of the GST disks can be written as the sum of partial contributions from these derived multipoles:

$$C_{scat} = \frac{\pi}{k^2} \sum_{\ell=1}^{\infty} \sum_{m=-\ell}^{l} (2\ell + 1)(|a(\ell, m)|^2 + |b(\ell, m)|^2). \tag{7}$$

We note that the above equations allow for calculating spherical multipoles of arbitrarily high order. As such, we can unambiguously identify not only fundamental but also higher-order multipoles of GST disks.

**Sample preparation**. A 280-nm-thick PMMA (950K AR-P 672.11) was spun onto CaF₂ substrate as an electron beam resist and baked on a hotplate for 3 min at 150 °C. Then a 50-nm-thick conductive protective coating (AR-PC 5090.02) was spun onto the PMMA film and baked for 2 min at 90 °C. This coating is used for the dissipation of e-beam charges on insulating substrates. The PMMA was exposed to define a nanohole array by E-beam lithography. All e-beam patterning was performed by SEM, which is equipped with a Raith Elphy Quantum lithography

system. The conductive layer was dissolved in DI water for 1 min and then the PMMA was developed in the developer (AR 600-56) for 3 min followed by rinsing in IPA. After the development, a 220-nm-thick GST film was then deposited onto the sample by magnetic sputtering with 50 W DC sputtering power while the substrate temperature was kept at room temperature. The deposited GST thin film was in its amorphous phase. The GST nanodisk array was realized after lift-off by ultra-sonic processing in acetone for 1 min.

**Sample characterization**. Transmission spectra were measured by using an infrared microscope (Hyperion1000) coupled to a Fourier transform infrared spectrometer (FTIR, Vertex 70). The detector used in the measurement is an MCT detector integrated into the microscope. The size of the probed area is $40 \times 40$ µm$^2$, corresponding to ~64 disks in the array with a 3 µm gap distance. All the measurements represent an average of 16 scans taken at a resolution of 4 cm$^{-1}$. To determine the transmittance ($T$) of the GST arrays, air was used as the reference. Experimental extinction spectra were then derived as $1 - T$. A 15× Schwarzschild objective is applied, which operates at ~16.7° off-normal to the surface of the sample and has a collection cone apex angle of ±7°. Given the small incident angle of the objective and the large diameter-to-thickness ratio of the GST disks, the fabricated structures in our study do not exhibit strong angular dispersion. Phase transformation of the GST material was induced by baking the samples on a hot plate which maintained a temperature of 145 °C. To ensure a systematic optical characterization of each sample, the phase-changing process and transmission measurement were implemented progressively. After being heated for 1 min, the sample was cooled down naturally and then transmission spectra were measured. Another cycle of the annealing process and the extinction measurement would be carried out thereafter until the GST disks were fully crystallized. The scanning electron microscope images were taken by Zeiss Ultra55 and the atomic force microscope (AFM) images were taken by VEECO Multimode.

## Data availability

The authors declare that all data supporting the findings of this study are available within this article and its supplementary information files or from the corresponding author upon reasonable request.

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

## Acknowledgements

J.T., H. L., Y.Q., and M.Q. acknowledge funding support from the National Key Research and Development Program of China (Grant No. 2017YFA0205700) and the National Natural Science Foundation of China (Grant No. 61425023). Y.Y., F.D., and S.I.B. acknowledge funding support from the European Research Council (the PLAQNAP project, Grant 341054) and the University of Southern Denmark (SDU2020 funding). J.T. and Y.Q. were also supported by Chinese Scholarship Council (CSC No. 201600160020, and No. 201706320254). D.Z. acknowledges support from the European Union's Horizon 2020 research and innovation program under the Marie SkÅ,odowska-Curie grant agreement no. 713683. Y.Y. thanks Dr. Kaikai Du for helpful discussions and Mr. Ziquan Xu and Dr. Vladimir Zenin for their great assistance in revising the manuscript.

## Author contributions

Y. Y. and D. Z. initiated the original idea of phase-change resonators. Y. Y conceived the presented concept and designed the experiment. J.T., Y.Y., and Y.Q. developed the theory and performed the computations. H.L. fabricated and characterized the samples. F.D. demonstrated the tunable beam steering of a GST metasurface. Y.Y. wrote the manuscript with J.T., and H. L., Y.Y., M.Q., and S.I.B. co-supervised the project and coordinated all the work. All the authors discussed the results and contributed to the final manuscript.

## Additional information

**Competing interests:** The authors declare no competing interests.

