## [Peer Review File · Nature Communications]

Reviewers' Comments:

Reviewer #1:

Remarks to the Author:

The work "Dynamic control of anapole states with phase-change alloys," by Tian, Luo, et al. studies tunable dielectric metasurfaces composed of a phase-change dielectric material, namely GST. The metasurface is designed such that it exhibits Mie-resonant modes in the mid-infrared spectral range, where the material absorption is negligible. In particular, the authors demonstrate the thermal tuning of the metasurface between its magnetic dipole mode and its anapole mode. This example is significant because the metasurface exhibits strong reflection at the magnetic dipole mode, while the scattering/reflection is completely suppressed at the spectral position of the anapole mode. Because of this qualitative change of the scattering of the resonators, one can achieve large intensity switching contrast. The authors demonstrate a contrast of 6 dB, which corresponds to about 75% of an intensity change.

The topic of tunable dielectric metasurfaces is a currently hot topic of interest, as reflected in the authors' reference list. Tuning of the resonator material or the surrounding environment is usually used for such tunable metasurfaces. The tuning of the resonator material is considered more useful, as the resonant mode is mostly confined inside the dielectric material. For such tuning, usually heating or carrier concentration modulation is traditionally implemented. The realization of tunable Mie-resonant dielectric metasurfaces by phase change materials is, therefore, a novel result. However, I believe that the fundamental concepts to reach this have already been explored previously (as explained below) and the results presented in current work represent only technical advances.

The three key concepts of the work by Tian, Luo et al. are: 1) the use of GST for tunable Mie-resonant metasurfaces; 2) Demonstrated switching from backward to zero scattering by temperature tuning; and 3) Demonstrated 75% of switching contrast. All three of these concepts have already been demonstrated earlier to some extent, as follows:

- 1) The approach to use dielectric phase change materials, such as GST for tunable dielectric metasurfaces was first pioneered by the group of Zheludev, in Ref.[48]. However, the work [48] did not explore any Mie-type resonances.
- 2) Switching between forward and backward scattering is not unique to the use of anapole mode. Such switching can be achieved when tuning between the magnetic resonance and the overlap mode of the electric and magnetic dipoles. Such switching from forward to backward scattering has already been demonstrated in Ref.[28]. However, the switching modulation was somewhat lower, about 20%.
- 3) High contrast intensity switching, of the order of 80% has also demonstrated with dielectric metasurfaces in APL 110, 071109 (2017).

The use of GST material has some additional advantages, which the authors discuss, however, do not demonstrate. For example:

- The change of the GST is non-volatile and reversible. However, the reversibility is not demonstrated. Tuning only in one direction has been explored. How do you switch back to the amorphous state? Can you demonstrate this? How many cycles can be performed if the process is reversible?
- Because the index change of the material is substantial, one can think of realizing a switchable device, e.g., a lens with a tunable focal length or a beam steering. Can the authors demonstrate a functional device?

Demonstrating the above functionalities would be truly original and would be worth publication in Nature Communications. However, with the current results, the originality of the work does not merit publication in this journal.

Technical comments:

- For their measurements, the authors use an FTIR microscope with reflective objectives, which

illuminate the metasurface under oblique incidence only. Please comment on the angular dispersion of the metasurface.

- The GST material is lossy even in the midIR wavelengths. Please comment on the effect of loss on the metasurface performance.

- The use of the notation "dark mode" to describe the anapole state is incorrect. Dark mode is used for a mode which is not excited by the incident light, e.g., the near-field intensity of the mode is zero. It is incorrect to use it to describe the far-field scattering, as the anapole mode is excited and the near-field is very high. The anapole mode is a bright mode, however, does not scatter in the far field. Please change your notations and remove the confusing use of "dark-mode."

Reviewer #2:

Remarks to the Author:

The manuscript „Dynamic control of anapole states with Phase-Change Alloys“ by J. Tian et al. reports on a thorough analysis and experimental proof of the dynamic tuning of various multipolar dielectric resonances including an anapole-type state by using the phase change material Ge₂Sb₂Te₅, short GST. The authors first give a nice theoretical explanation of the various bright and dark states by calculating the Mie-resonances of a spherical particle made from GST as a function of the level of crystallization. As expected for any dielectric material, also the GST shows a range of multipolar resonances which can be attributed to electric dipole (ED), magnetic dipole (MD), etc. and a minimum in scattering attributed to the anapole state. As the GST dramatically changes its dielectric function upon crystallization, also the corresponding resonances shift, which the authors nicely experimentally prove on GST nanodiscs with varying crystallization fractions and diameters. As both the ED and the anapole resonances do shift by the same amount upon the crystallization of the GST, an optical switch with multi-level tunability becomes possible.

Overall, this very well written manuscript is a timely contribution merging two “hot” topics in nanophotonics, dielectric resonators and the tunability introduced by phase change materials. As there is a wealth of different modes possible, I believe that this manuscript will stimulate further research and open up a new field for active dielectric photonics. I thus can clearly recommend it for publication in Nature Communications after a few open questions are answered.

Minor Comments:

1) When looking at the dielectric data of GST-225 (Fig. S1) , it's losses in the crystalline state ($k = 0.2$) should be much higher compared to an almost lossless dielectric as e.g. Si. Thus I am wondering why the resonances in the experimental and simulated spectra (Figs. 1, 3, 4) are not so much broader in the crystalline state? In Fig. 1, at least the MD state seems to be much broader and weaker for 100% crystallization while the ED state does not seem to change a lot. A better discussion on the losses in the crystalline GST and their (different?) influence on the various modes would be helpful for the reader and could stimulate even further research.

2) On a similar note, an example for optimizing the GST stoichiometry for photonic applications in the desired spectral range can be found in [A.-K. U. Michel et al., Advanced Optical Materials 2017, 5, 1700261.], which should also be cited as it clearly states the origin of losses in c-GST (Bandgap absorption and free carriers).

3) In Figure 3, I do not understand why the experimental spectra are much sharper (more pronounced & deeper minima for the A1 mode) compared to the simulations? Usually, the experiments have additional sources for losses compared to the simulation, but here it seems to be the opposite. A discussion on why this is the case should be added.

4) For both refs. 6 & 36, the page or article numbers look strange to me.

5) The title talks about "phase-change alloys", although only a single Phase-change alloy, GST-225, is used.

Reviewer #3:

Remarks to the Author:

J. Tian et al. discuss a numerical and experimental study of the phase-change material (PCM) Ge₂Sb₂Te₅ (GST-225) nanodisks supporting scattering bright and dark modes in the infrared spectral range. Overall, the manuscript is written very clearly and the figures help the reader to understand their study. The study of the shifting between the anapole and the electric dipole modes is novel and a smart approach to make use of the refractive index contrast of the PCM. However, the manuscript has to be improved in terms of connecting the results to literature, the possibility to reproduce the results and the appropriate use of precise terms. In the following, these aspects will be described in more detail (The order is not related to the importance.):

1. The authors missed out on relating their work with the work of Y. G. Chen, S. A. Maier, M. H. Hong et al. published in 2013 in Optics Express (DOI: 10.1364/OE.21.013691). In this publication, the authors study Au nanodisks covered by a GST-225 thin film and heated the sample for more than 60 min at 135°C to realize different fractions of crystallinity in the PCM film, which allowed for stepwise switching of the LSPR. The submitted study should not only cite this publication, but also relate their own results to this work.

2. Furthermore, the authors missed out on commenting on the targeted spectral range. The same holds for the picked phase-change material. No reasoning for this choice was given in the manuscript. The PCM selection regarding different GST compounds was discussed in Advanced Optical Materials last year by A. U. Michel, M. Wuttig and T. Taubner (DOI: 10.1002/adom.201700261). The authors should add a brief related discussion in the manuscript.

3. Additionally, it does not seem obvious why to choose the Lorentz-Lorenz effective medium theory over other approaches, such as e.g. Maxwell-Garnett. The authors should discuss the choice they made for the submitted study.

4. The authors should discuss more clearly, how they determine the crystallinity C as a function of the annealing time. To me, this aspect is missing and crucial for the validity of the presented results. A SEM image as well as an AFM scan of the nanodisks after different annealing times should be added and could support the presented evaluation.

5. Overall, the title of the submitted manuscript has to be reviewed since it connotes a different content to the reader meaning that one expects results including reversible switching by the means of optical or electrical pulses.

6. The terms "tunability", "switchability" and "reversibility" should be clearly defined, carefully used and not be mixed up in the manuscript to avoid any misguidance of the reader.

- The variation of crystallinity C can be interpreted as tunability, whereas switchability does not seem to be suited in terms of "mode switching". In this sense, a control via optical or electrical pulses can be easily associated, which is misleading. "Mode shifting" might be a better suited term.

- The term "reversibility" should be used carefully as well. In the presented work, the crystallinity is changed via annealing, which is to some extent a controllable and reproducible process. This approach does not allow for targeting any desired C , i.e. only an increase of C is possible, which is in contrast to optical or electrical switching. In the given geometry, a reversible targeting of C between 0 and 100% with optical or electrical pulses is not straightforward and should be discussed here. This aspect is missing and thus, the authors promise more functionality than what

could be realized with the presented layout. The outlook towards metadevices or reversible cycling of the GST-225 structural phase even, needs to take into account, that many changes on the sample layout would be necessary to realize reversibility. These adaptations might hinder the detection of the anapole modes. The authors need to take this into account to not misguide the reader.

7. Please consider the following technical comments as suggestions towards a complete technical description and an easily readable publication:

- What is the size of the probed sample area (number of disks in the array, FTIR microscope aperture size)? Moreover, more FTIR measurement details are missing, which are crucial for reproducing and rating the presented results, such as background normalization, averaged number of scans, resolution and applied objective. Please add these details to the Methods section.
- The MQ and EQ modes can not (easily) be seen in figures 1 and S2. Either leave them out or mark them in the figures.
- Please rephrase or elaborate on the term “[...] with any crystalline phases of GST.” used on page 6.
- Please comment on the term “dispersionless tunability” on the top of page 8.
- Please add a reference to the refractive index of CaF₂ on page 13.
- Please correct the caption of figure S3: “[...] We mention that the Fig. S3c is the same [...].”
- Typo on page 14: “Vertex70”
- Typo on page 15: “Cambridge University Press”

In conclusion, I recommend a resubmission after major revisions.

Response to Reviewer #1

General comment 1) *The work "Dynamic control of anapole states with phase-change alloys," by Tian, Luo, et al. studies tunable dielectric metasurfaces composed of a phase-change dielectric material, namely GST. The metasurface is designed such that it exhibits Mie-resonant modes in the mid-infrared spectral range, where the material absorption is negligible. In particular, the authors demonstrate the thermal tuning of the metasurface between its magnetic dipole mode and its anapole mode. This example is significant because the metasurface exhibits strong reflection at the magnetic dipole mode, while the scattering/reflection is completely suppressed at the spectral position of the anapole mode. Because of this qualitative change of the scattering of the resonators, one can achieve large intensity switching contrast. The authors demonstrate a contrast of 6 dB, which corresponds to about 75% of an intensity change.*

The topic of tunable dielectric metasurfaces is a currently hot topic of interest, as reflected in the authors' reference list. Tuning of the resonator material or the surrounding environment is usually used for such tunable metasurfaces. The tuning of the resonator material is considered more useful, as the resonant mode is mostly confined inside the dielectric material. For such tuning, usually heating or carrier concentration modulation is traditionally implemented. The realization of tunable Mie-resonant dielectric metasurfaces by phase change materials is, therefore, a novel result.

However, I believe that the fundamental concepts to reach this have already been explored previously (as explained below) and the results presented in current work represent only technical advances.

The three key concepts of the work by Tian, Luo et al. are: 1) the use of GST for tunable Mie-resonant metasurfaces; 2) Demonstrated switching from backward to zero scattering by temperature tuning; and 3) Demonstrated 75% of switching contrast. All three of these concepts have already been demonstrated earlier to some extent, as follows:

1) The approach to use dielectric phase change materials, such as GST for tunable dielectric metasurfaces was first pioneered by the group of Zheludev, in Ref.[48]. However, the work [48] did not explore any Mie-type resonances.

2) Switching between forward and backward scattering is not unique to the use of anapole mode. Such switching can be achieved when tuning between the magnetic resonance and the overlap mode of the electric and magnetic dipoles. Such switching from forward to backward scattering has already been demonstrated in Ref.[28]. However, the switching modulation was somewhat lower, about 20%.

3) High contrast intensity switching, of the order of 80% has also demonstrated with dielectric metasurfaces in APL 110, 071109 (2017).

Our response: We thank the reviewer for his/her constructive comments on the novelty of our study. First of all, we would like to appreciate the reviewer for pointing out the novelty of our research by using phase-change materials to construct Mie-resonant metasurfaces. Regarding the references and comparisons provided by the reviewer, we have carefully examined all of them, and believe that our present work has rather significant differences with all the previous reports in terms of physical mechanisms involved and device performance demonstrated:

- 1) Our work reports the first experimental demonstration of actively controlled Mie resonances in *standalone* GST nanostructures and arrays. In Ref. 52 (original 48), Wang *et al.* utilized laser-induced phase changes to write different patterns in GST thin films (matrix). The optical contrast ($\Delta n = n_c - n_a$) between the written patterns (crystalline) and the background (amorphous matrix) is around 1 in the visible or 2 in the mid-infrared range. It is worth noting that this level of optical contrast is not sufficient to support strong Mie resonances: strong Mie resonances require sufficiently small scattering volumes (to keep damping by scattering relatively weak), implying the use of very large optical contrast (the size of Mie resonant particles in air is of the order of the light wavelength in the particle medium). The Fig. 5 in Ref. 52 further confirms this argument, where the reflection and transmission spectra do not show a strong resonant feature (merely 5% intensity change at the resonance wavelength of $\sim 2 \mu\text{m}$), even with closely packed unit cells. By contrast, we demonstrate that a standalone GST nanostructure ($n_a > 4$ and $n_c > 6$) even in its simplest geometry (sphere) can support a distinct series of strong Mie resonances including the magnetic-type, the electric-type, and the anapole states with an extra possibility for active control and mode shifting. We note that, for the same reason, high index contrast between the dielectrics and the surroundings is the prerequisite in meta-optics, where Mie resonances and associated multipolar effects lie at the heart of the field and provide the physical foundations for almost all the applications including directional scattering, metasurfaces, harmonic generation, sensing, and topological photonics, etc. [Ref. 5-9]. Therefore, we believe that a thorough exploration of the Mie resonances in standalone phase-change nanostructures is unique and of high importance to the community, as it uncovers an entirely new direction to realize dielectric devices with active tunability.
- 2) We agree with the reviewer that the switching between forward and backward scattering does not necessarily require the use of anapole states and can be also realized by other mechanisms. In Ref. 30 (original 28), Makarov *et al.* experimentally demonstrated a 20% tuning of the reflectance of a Si nanosphere by photoinjection of electron-hole plasma, an effect that is weaker than that in our study (75%), as justly pointed out by the reviewer. At the same time, we would also like to mention that, demonstration of a multispectral optical switch using anapole states is only one representative example and application of our results. As was discussed above, our study shows a diverse set of Mie resonances supported by GST nanostructures, which hold promise for various possibilities. In the revised manuscript, we have added two additional numerical studies to exemplify other possibilities and potential applications of our presented results: i) tunable scattering directionality and mode shifting between magnetic and electric dipole resonances in single GST nanoparticles, and ii) tunable beam steering in GST disk arrays. We hope that our revision and additional examples would help to properly convey the importance and impact of our work.

Meanwhile, the optical anapole itself is an intriguing physical phenomenon that has been discovered only in recent years [Ref. 13]. Our study reports the first active use of optical anapoles and shows their direct application by demonstrating an optical switch. Given the great potential of optical anapoles in many other scenarios such as optical nonlinearity, biosensing, enhanced Raman scattering, and nanolasers [see Ref. 13–25], the demonstrated active control in our study involving this remarkable phenomenon is naturally expected to find other applications and/or trigger future developments besides the presented optical switch.

- 3) We thank the reviewer for suggesting a relevant paper, and it has now been cited as Ref. 32 in the main text. In Ref. 32, Komar *et. al.* demonstrated tunable dielectric metasurfaces based on liquid crystals. As discussed in the introduction of our paper and also mentioned by the reviewer, such a method (tuning the refractive index of the surrounding environment) is usually not as efficient as tuning the resonator material index itself. Therefore, the physical mechanisms and device performance in Ref. 32 are entirely different from those in our study. For instance, whereas the transmission modulation reported in Ref. 32 is slightly higher than our results, it has a very limited tunable spectral range (~ 50 nm at the resonance wavelength of ~ 1550 nm, corresponding to a 3% fractional bandwidth). In strong contrast to that, our study reports a much broader tuning range (e.g. for A1 mode, the tuning range is around $1.5 \mu\text{m}$ at the resonance wavelength of $\sim 4 \mu\text{m}$, corresponding to the fractional bandwidth $> 30\%$). This substantial increase in the bandwidth allows us to actively control multiple mode shifting, which is not possible with the previous approach. Furthermore, the utilization of anapole states makes our structure ($\sim \lambda/20$) much thinner than that in Ref. 32 ($\sim \lambda/7$, even disregarding the $5\text{-}\mu\text{m}$ -thick liquid crystal).

To summarize, we present the first experimental study of actively tunable Mie resonances in the *structured* phase-change material GST. By applying a rigorous multipole analysis, we systematically investigate the broadband tuning of distinct Mie resonances and associated multipolar effects in GST nanostructures. The obtained results are expected to open up an entirely new direction for realizing active nanophotonic devices and stimulate a variety of further developments. Several examples are provided to illustrate the potential applications of our findings. In particular, we focus ourselves on the active control of anapole states in GST nanodisks and demonstrate, as a representative example, an ultrathin ($\lambda/20$), multispectral optical switch with multi-level capability and a high contrast ratio $\sim 75\%$. Compared to the previous reports, our study provides substantially new physical insights leading to dramatically different device performance. Therefore, we believe that our work reports a timely and significant progress in the field of nanophotonics beyond technical advances.

In the revised supplementary information, we have added the following two demonstrations to substantiate the importance and originality of our findings:

Supplementary Note 3: Tunable scattering directionality of single GST nanospheres

The broadband active tuning of the Mie resonances in the structured GST can lead to a variety of interesting phenomena. Here we exemplify this point by considering the same GST nanosphere ($R = 450$ nm) as in Fig. 1. The scattering spectra of the GST sphere with three different crystallinities are plotted in Supplementary Fig. 4a, showing the investigated wavelength λ_c at $3.97 \mu\text{m}$ with a dotted line. To depict the far-field scattering patterns, the same coordinate as in Fig.1 is adopted, in which the incident wave propagates along the x -axis with the polarization of the electric field along the z -axis.

When the GST sphere is at the amorphous state, it supports a magnetic dipole resonance at λ_c . The far-field scattering in Supplementary Fig. 4b shows a typical radiation pattern of a magnetic dipole oriented along the y -axis. By contrast, after introducing a moderate phase change of 25%, the scattering spectrum shows an intersection between the electric and magnetic dipole contributions. The spectral overlap and equal far-field strengths of the two dipoles indicate the satisfaction of the second Kerker condition [S6]., as confirmed by the unidirectional scattering in the backward direction. When the phase change continues increasing, the sphere finally reaches its crystalline

state with its scattering similar to that of a typical electric dipole oriented along the z -axis, i.e., the scattering pattern (Supplementary Fig. 4d) transforms in orthogonal to that of the amorphous sphere. Therefore, mode shifting between magnetic and electric dipole resonances could also be realized with the GST sphere, which may make a fundamental impact on many intriguing physical phenomena related to Mie resonances.

Supplementary Figure 4. (a) Progressive multipole decomposition of the scattering spectra of the GST sphere in Fig. 1. (b-d) Far-field scattering patterns of the GST sphere at $\lambda_c = 3.97 \mu\text{m}$ with three different crystallinities, i.e. $C = 0\%$, 25% , and 100% .

Supplementary Note 7: Tunable beam steering of a GST metasurface

In addition to the active control of anapole states and presented optical switch in the main text, here we numerically demonstrate an alternative application of GST metasurfaces composed of disk arrays for tunable and efficient beam steering.

Supplementary Figure 10. (a) The cross sections of a unit cell in the tunable metasurface consisting of GST disks. (b) Illustration of a super cell consisting of 4 GST disks with varied diameter. To realize the demonstrated functionality, here the diameters D_1 to D_4 are 770 nm , 1130 nm , 1310 nm , and 1340 nm , respectively.

Following the general principle for designing gradient metasurfaces [S9], here we first set the height of the GST disks to 600 nm and the period of a unit cell to 1800 nm. By varying the diameter of the disks, the Mie resonances supported by the disks would undergo spectral shifts and thereby exhibit varied phase response at the design wavelength $\lambda_d = 4 \mu\text{m}$. In this way, we can introduce a linear phase gradient along the x -direction, parallel to the incident polarization, as seen in Supplementary Fig. 10. According to the generalized Snell's law [S10], the incident light would be anomalously transmitted into a specific angle, as shown in the Supplementary Fig. 11a. 87.6% of the transmission is propagating along the +1 diffraction order at λ_d with light in other diffraction orders being strongly suppressed. By contrast, when we introduce a 30% phase change in the GST disks, the refractive index of GST would increase and lead to dramatic redshifts of the supported resonances, thereby limiting the phase variation at λ_d . As such, the GST disk arrays would support a nearly constant phase response along the interface, resulting in the metasurface exhibiting the conventional (zero-order) transmission (Supplementary Fig. 11b) with most of the light (95.4%) propagating normally. Hence, in this manner, one can realize a GST metasurface with tunable beam steering by utilizing a simple configuration of GST disks.

Supplementary Figure 11. (a) Transmission spectra (left) for different diffraction orders ($|m| \ll 1$) of the GST metasurface composed of amorphous GST disks. The calculated electric field distribution (right) at $\lambda_d = 4 \mu\text{m}$ showing anomalous transmission. (b) Transmission spectra (left) for different diffraction orders ($|m| \ll 1$) of the GST metasurface composed of 30% cGST disks. The calculated electric field distribution (right) at $\lambda_d = 4 \mu\text{m}$ showing ordinary (zero-order) transmission along the $+z$ direction.

Technical Comment 2) *The use of GST material has some additional advantages, which the authors discuss, however, do not demonstrate. For example:*

2a) *The change of the GST is non-volatile and reversible. However, the reversibility is not demonstrated. Tuning only in one direction has been explored. How do you switch back to the*

amorphous state? Can you demonstrate this? How many circles can be performed if the process is reversible?

Our response: We agree with the reviewer that GST has a variety of advantages such as dramatic optical contrast, non-volatility, and reversibility, which are exactly the reasons why we are so interested in this promising material and carried out the study. However, most of these advantages especially the reversible tuning of GST has only been reported in the configurations of thin films [Ref. 39–50]. To the best of our knowledge, our work is the first systematic and experimental investigation on the Mie resonances in individual GST nanoparticles or metasurfaces. Given the several detailed demonstrations of active multipolar effects in 3D GST nanostructures, we believe our current results are sufficient to stimulate further developments and research interest along this direction. A thorough experimental demonstration of reversible tuning in 3D GST nanostructures would be a significant development, but this formidable challenge is beyond the scope of our current study.

[Redacted]

2b) *Because the index change of the material is substantial, one can think of realizing a switchable device, e.g., a lens with a tunable focal length or a beam steering. Can the authors demonstrate a functional device?*

Demonstrating the above functionalities would be truly original and would be worth publication in Nature Communications. However, with the current results, the originality of the work does not merit publication in this journal.

Our response: We thank the reviewer for the suggestion. Such a tunable and functional device is indeed a significant way to utilize our current findings. To exemplify these possibilities, we have added a numerical demonstration (Supplementary Note 7) of a GST metasurface with tunable beam steering. The device is composed of GST disk arrays with varying diameters within a supercell, thereby constructing a gradient metasurface with active tunability. For GST disks at the amorphous state, the metasurface supports a linear phase gradient resulting in efficient anomalous transmission, with 87.6% of the transmitted light propagating along the +1 diffraction order. By contrast, after introducing a 30% phase change, the metasurface acquires a nearly constant phase response along the interface that enforces 95.4% of the transmitted light to propagate normally.

In addition to the GST metasurface, we have also discussed how to exploit our current findings in the situation of individual GST nanoparticles (Supplementary Note 3). Besides the active control of anapole states, we further show the possibility to tune the scattering directionality of a single GST sphere and achieve mode shifting between the magnetic and electric dipole resonances. All these results are original and expected to make significant impacts on many fundamental physical phenomena and technological applications that are directly related to Mie resonances [Ref. 5-9]. With these additional demonstrations, we hope that we have exemplified the importance and originality of our findings more clearly and convincingly. Altogether, we believe that our work reports a significant progress in the field of nanophotonics and now meet the high standards of *Nature Communications*.

The mentioned demonstrations and supplementary notes have been presented in the response to the previous comments. In the main text, we have also referred to these two examples as:

We note that, given the plethora of phenomena caused by Mie resonances and associated multipolar effects [5-9], active tunability can straightforwardly be implemented into many existing applications by utilizing GST resonators. For instance, in Supplementary Note 3 and Supplementary Fig. 4, we explore the possibilities to realize the mode shifting between the MD and ED resonances. Such a tunability also allows us to actively control the directionality of corresponding far-field scattering.

Besides, similar to the case of GST spheres, other mode shifting effects could also be realized by exploiting intermediate phases in the GST disk arrays with properly designed geometry, a new fascinating direction that promises many future developments. In Supplementary Note 7 (and Supplementary Figs. 10 and 11), we exemplify these possibilities by providing a numerical demonstration of a GST metasurface for tunable beam steering.

Technical comment 3) *For their measurements, the authors use an FTIR microscope with reflective objectives, which illuminate the metasurface under oblique incidence only. Please comment on the angular dispersion of the metasurface.*

Our response: The applied objective in our study operates at an off-normal angle of 16.7° and we have added the details in the Method section as:

A $15\times$ Schwarzschild objective is applied, which operates at $\sim 16.7^\circ$ off-normal to the surface of the sample and has a collection cone apex angle of $\pm 7^\circ$.

Regarding the angular dispersion of the disk arrays, we have simulated the scattering spectra of GST disks (Height = 220 nm, Diameter = $2\ \mu\text{m}$, the same as in Figure 3 in the main text) at varied incident angles θ :

Figure R3. TE (upper) and TM (bottom) excitation of the GST nanodisk in Figure 3 with an incident angle θ and corresponding scattering spectra for aGST (left), 50%-cGST (middle), and cGST (right), respectively.

As can be seen, for both TE and TM illumination, the spectral positions of the Mie resonances are generally insensitive to the incident angle below 30° , and the anapole states keep their significant feature of suppressed far-field scattering. Therefore, the effects of mode shifting are maintained with varied incident angle $\theta < 30^\circ$. The angle insensitivity can be attributed to the relatively small thickness ($\sim \lambda/20$) of the GST disks used in our study. Given the small incident angle with the objective applied in our experiment, our metasurface thus is not expected to exhibit strong angular dispersion.

Technical comment 4) *The GST material is lossy even in the midIR wavelengths. Please comment on the effect of loss on the metasurface performance.*

Our response: In the revised manuscript, we have added extensive discussions in Supplementary Note 2 and Supplementary Note 6 to investigate the impacts of the material loss on the optical response of GST nanostructures.

Technical comment 5) *The use of the notation "dark mode" to describe the anapole state is incorrect. Dark mode is used for a mode which is not excited by the incident light, e.g., the near-field intensity of the mode is zero. It is incorrect to use it to describe the far-field scattering, as the anapole mode is excited and the near-field is very high. The anapole mode is a bright mode, however, does not scatter in the far field. Please change your notations and remove the confusing use of "dark-mode."*

Our response: We thank the reviewer for the correction. We have now removed the misuse of "dark mode" throughout our manuscript and replaced it with the term "scattering dark state" [*Nano Lett.*, **14**, 2783-2788, (2014) and *Light Sci. Appl.*, **6**, e16197, (2017)] to describe the suppressed far-field scattering of optical anapoles. The relevant descriptions in our manuscript have been revised as follows:

Such an intricate field distribution is characterized by a significant suppression of far-field scattering. Therefore, this phenomenon is also commonly referred to as a scattering dark state.

In particular, we then demonstrated the shifting between ED and anapole modes, corresponding to scattering bright and dark states, respectively.

Response to Reviewer #2

General comment. The manuscript „Dynamic control of anapole states with Phase-Change Alloys“ by J. Tian et al. reports on a thorough analysis and experimental proof of the dynamic tuning of various multipolar dielectric resonances including an anapole-type state by using the phase change material Ge₂Sb₂Te₅, short GST. The authors first give a nice theoretical explanation of the various bright and dark states by calculating the Mie-resonances of a spherical particle made from GST as a function of the level of crystallization. As expected for any dielectric material, also the GST shows a range of multipolar resonances which can be attributed to electric dipole (ED), magnetic dipole (MD), etc. and a minimum in scattering attributed to the anapole state. As the GST dramatically changes its dielectric function upon crystallization, also the corresponding resonances shift, which the authors nicely experimentally prove on GST nanodiscs with varying crystallization fractions and diameters. As both the ED and the anapole resonances do shift by the same amount upon the crystallization of the GST, an optical switch with multi-level tunability becomes possible.

Overall, this very well written manuscript is a timely contribution merging two “hot” topics in nanophotonics, dielectric resonators and the tunability introduced by phase change materials. As there is a wealth of different modes possible, I believe that this manuscript will stimulate further research and open up a new field for active dielectric photonics. I thus can clearly recommend it for publication in Nature Communications after a few open questions are answered.

Our response: We thank the reviewer for his/her positive comments and clear support for our work.

Technical comment 1) When looking at the dielectric data of GST-225 (Fig. S1) , it's losses in the crystalline state ($k = 0.2$) should be much higher compared to an almost lossless dielectric as e.g. Si. Thus I am wondering why the resonances in the experimental and simulated spectra (Figs. 1, 3, 4) are not so much broader in the crystalline state? In Fig. 1, at least the MD state seems to be much broader and weaker for 100% crystallization while the ED state does not seem to change a lot. A better discussion on the losses in the crystalline GST and their (different?) influence on the various modes would be helpful for the reader and could stimulate even further research.

Our response: We thank the reviewer for the insightful comment. To address the reviewer's question, we have carefully examined the influence of crystallinity C (and associated losses) on the quality factors (Q factors) of ED and MD responses in a single GST sphere ($R = 450$ nm, the same as in Fig. 1).

Figure R4. (a) Normalized scattering partial contribution of ED (a_1) in a single GST nanosphere with $R = 450$ nm at different crystallinities C . The shaded area indicates the spectral region close to an anapole state and used for Fano fitting. (b) Fano fitting of the asymmetric line shape at the anapole state in a GST nanosphere.

We plot the partial scattering of ED with respect to different crystallinities in Figure R4a. We can observe that the ED scattering features an evident Fano line shape due to the interference between a resonant eigenmode (internal) and non-resonant background pathway (external) [*Nat. Photon.*, **11**, 543 – 554 (2017)]. To extract the Q-factor, the partial scattering contribution of ED at the anapole states was fitted into a Fano line shape given by [*Nat. Mater.*, **13**, 471 – 475 (2014)]:

$$\sigma_{\text{ED}}(\omega) \propto \frac{\left(q \frac{\Gamma}{2} + \omega - \omega_0\right)^2}{\left(\frac{\Gamma}{2}\right)^2 + (\omega - \omega_0)^2}, \quad (1)$$

with ω_0 is the central resonant frequency; Γ is the full-width at half-maximum (FWHM) of the resonance; q is the asymmetry parameter describing the ratio between the resonant scattering and the non-resonant background. An excellent agreement between the Mie calculation and the Fano fit can be seen in Figure R4b. For all the crystallinities, the asymmetry parameter q is close to 1, indicating the resonant and the non-resonant pathways have similar amplitudes. The Q-factor was then determined by $Q = \omega_0/\Gamma$, as plotted in Figure R5.

Figure R5. Relation between the Q-factor of the anapole states with respect to different crystallinities.

From Figure R5, we can quantitatively conclude that the ED scattering does not exhibit dramatic broadening linewidths for increasing crystallinities. This is because a larger crystallinity in GST would bring in increases in both real and imaginary parts of the refractive index $n = n_0 + ik$. For the ED resonance, a larger n_0 would make the structure a more perfect scatterer with a smaller radiative damping and thus lead to a higher Q-factor [*Phys. Rev. A*, **93**, 053837, 2016]. Meanwhile, a larger k would result in larger dissipative damping with a smaller Q-factor. Such a trade-off explains why all the ED resonances in Figs. 1, 3 and 4 do not show substantial linewidth broadening and also accounts for the appearance of the maximized Q-factor at $C = 50\%$.

Figure R6. (a) Normalized partial scattering contribution of MD (b₁) in a single GST nanosphere with $R = 450$ nm at different crystallinities C . (b) Q-factor of MD contribution with respect to crystallinities C .

In contrast to the asymmetric Fano line shape of the ED contribution, the MD response manifests a symmetric Lorentzian line shape, as shown in Figure R6a. This is because the internal resonance arising from the circular displacement currents at the MD state is much stronger than the background pathway, thereby dominating in the interference with $q \gg 1$ in Eq. (1). Thus, we can directly obtain the FWHM from the spectra and determine the Q-factor of the MD resonances, as shown in Figure R6b. A clear decline in the Q-factor with increasing crystallinities could be seen. We attribute this response to the strong resonant feature at MD states and its large field concentration inside the particle (see Fig. 1d). As such, the increase in k , which significantly decreases the Q-factor, would have a much larger impact on the linewidth than the increase in n_0 . In the revised manuscript, we have added the above discussions in Supplementary Note 2 and refer it in the main text as:

...in Supplementary Fig. 2a we provide a detailed multipole analysis for other crystalline phases of GST, clarifying that the progressive shifts of the multipolar effects indeed take place in the GST sphere with all different crystallinities. The effects of the crystallinity and material loss on the bandwidths of different Mie states are also discussed in Supplementary Note 2 and shown in Supplementary Fig. 3.

Technical comment 2) On a similar note, an example for optimizing the GST stoichiometry for photonic applications in the desired spectral range can be found in [A.-K. U. Michel et al., *Advanced Optical Materials* 2017, 5, 1700261.], which should also be cited as it clearly states the origin of losses in c-GST (Bandgap absorption and free carriers).

Our response: We thank the reviewer for recommending the important reference and now it has been cited in the manuscript as Ref. 57. We have added a few lines in the main text to relate the reason why we choose the targeted spectral range to the material loss of GST:

Here we focus on the mid-infrared range given the high refractive index, substantial optical contrast, and relatively low loss of the GST material [57].

Technical comment 3) In Figure 3, I do not understand why the experimental spectra are much sharper (more pronounced & deeper minima for the A1 mode) compared to the simulations? Usually, the experiments have additional sources for losses compared to the simulation, but here it seems to be the opposite. A discussion on why this is the case should be added.

Our response: We thank the reviewer for the comment. To address the reviewer's concern, we have examined this point carefully by directly comparing the experimental and simulation spectra as shown in Figure R7.

Figure R7. Comparison between normalized simulation and experimental extinction spectra (pitch size = 5 μm) for GST nanodisks with diameter $D = 2 \mu\text{m}$, height $H = 220 \text{ nm}$ in amorphous (a) and crystalline (b) states.

For aGST, where there is no absorption loss, the simulated resonance spectrum is in fact sharper than the experimental one. For cGST, the simulated spectrum features resonances with similar (to the experimental) bandwidths, but the resonance seems to be less pronounced at the first-order anapole state. This slight difference might be attributed to several reasons, for example, to the applied objective operating at small incident and collection angles, the lattice effect in the GST arrays (although negligible to the spectral position but may influence the resonance strength), or to a slight overestimation of the material loss used in the simulation, etc. To provide a more quantitative comparison, we applied the same technique in Figure R4 to R6 to analyze the Q-factor of the anapole states in the simulations and experiment. The results are shown in Figure. R8.

Figure R8. Q-factor of anapole states in simulation (black circles) and experiment (red triangles) at different crystallinities.

As can be seen from the above figure, the Q-factors of the first-order anapole state in both simulation and experiment generally agree well with each other. Moreover, both of them do not show dramatic changes with increasing crystallinity C , similar to what we have found in GST spheres (Figure R5).

In the revised manuscript, we have added a few lines in the main text to discuss the reasons accounting for the slight mismatch between the simulation and experimental spectra:

Slight deviations in relative resonance strengths and linewidths might be attributed to the non-zero (although small) oblique incident angle provided by the applied objective, lattice effect, or a slight overestimation of the material loss used in the simulation.

Technical comment 4) For both refs. 6 & 36, the page or article numbers look strange to me.

Our response: We thank the reviewer for the thoughtful comment. We have double-checked and corrected the format of all the references in our manuscript.

Technical comment 5) The title talks about "phase-change alloys", although only a single Phase-change alloy, GST-225, is used.

Our response: We thank the reviewer for the correction. We have revised the title as “Active control of anapole states by structuring the phase-change alloy $\text{Ge}_2\text{Sb}_2\text{Te}_5$ ”, in which we have now specified the name of the phase-change material that is used in our study.

Response to Reviewer #3

General comment. *J. Tian et al. discuss a numerical and experimental study of the phase-change material (PCM) Ge₂Sb₂Te₅ (GST-225) nanodisks supporting scattering bright and dark modes in the infrared spectral range. Overall, the manuscript is written very clearly and the figures help the reader to understand their study. The study of the shifting between the anapole and the electric dipole modes is novel and a smart approach to make use of the refractive index contrast of the PCM.*

However, the manuscript has to be improved in terms of connecting the results to literature, the possibility to reproduce the results and the appropriate use of precise terms. In the following, these aspects will be described in more detail (The order is not related to the importance.):

Our response: We thank the reviewer for his/her positive evaluation of the novelty of our work. To address the reviewer's concerns, we have substantially revised the manuscript following the reviewer's comments.

Technical comment 1) *The authors missed out on relating their work with the work of Y. G. Chen, S. A. Maier, M. H. Hong et al. published in 2013 in Optics Express (DOI: 10.1364/OE.21.013691). In this publication, the authors study Au nanodisks covered by a GST-225 thin film and heated the sample for more than 60 min at 135°C to realize different fractions of crystallinity in the PCM film, which allowed for stepwise switching of the LSPR. The submitted study should not only cite this publication, but also relate their own results to this work.*

Our response: We thank the reviewer for mentioning the important literature and now we have highlighted this work in our introduction as:

Chen et al. realized stepwise tuning of the lattice resonance in a hybrid plasmonic system consisting of a gold disk array and an underneath GST thin film [40].

We also found that we have used the same technique as in the Ref. 40 to estimate the crystallinity C of GST, therefore we have related our results to this work as:

Here we estimate the crystallinity C by matching the spectral position of the anapole states in the experiment, similar to Ref. [40].

Technical comment 2) *Furthermore, the authors missed out on commenting on the targeted spectral range. The same holds for the picked phase-change material. No reasoning for this choice was given in the manuscript. The PCM selection regarding different GST compounds was discussed in Advanced Optical Materials last year by A. U. Michel, M. Wuttig and T. Taubner (DOI: 10.1002/adom.201700261). The authors should add a brief related discussion in the manuscript.*

Our response: We thank the reviewer for suggesting the relevant paper and now it has been cited as Ref. 40. We have also added the following discussions about our choice of targeted spectral range and the material in the main text:

In fact, among all the phase-change chalcogenides, GST features one of the highest refractive indices in its amorphous states [57], which satisfies the essential prerequisite for constructing dielectric nanoantennas with strong Mie resonances.

Here we focus on the mid-infrared range given the high refractive index, substantial optical contrast, and relatively low loss of the GST material [57].

As a final remark, we mention that there is a wide selection of phase-change chalcogenides featuring extraordinary optical contrasts and low loss [57], holding the promise for further developments and other opportunities.

Technical comment 3) *Additionally, it does not seem obvious why to choose the Lorentz-Lorenz effective medium theory over other approaches, such as e.g. Maxwell-Garnett. The authors should discuss the choice they made for the submitted study.*

Our response: We have chosen the Lorentz-Lorenz relation out of various effective medium theories (EMTs) because it is so far the most widely used approach in the simulation of hybrid or isolated GST nanostructures [e.g. Ref. 40 and 56]. In addition, we have also compared the refractive indices of semicrystalline GST estimated by Lorentz-Lorenz and Maxwell-Garnett mixing rules [*Appl. Opt.*, **46**, 4065 – 4072 (2007)], as shown in Figure. R9. We can clearly see that, the maximum difference in the real part of the refractive index n_0 calculated by these two approaches is smaller than 5% for all the crystallinities. Therefore, we can safely conclude that our estimation using the Lorentz-Lorenz relation should be well within the acceptance of EMT approximation.

Figure R9. (a) Comparison between the real part of refractive indices n_0 of semicrystalline GST estimated by Lorentz-Lorenz (solid line) and Maxwell-Garnett (dashed line) mixing formulae. Five different colors represent five different crystallinities C (10%, 30%, 50%, 70%, and 90%). (b) The maximum difference in n_0 calculated by the two approaches was a function of the crystallinity C .

In the revised manuscript, we have added the following lines in the Methods Section to discuss our choice of the Lorentz-Lorenz relation:

For any intermediate phases with a crystallinity C ($0 \leq C \leq 1$), the dielectric constant $\epsilon_{\text{GST}}(\lambda, C)$ can be estimated by using the effective medium theories (EMTs). Out of various EMTs, here we chose to use the Lorentz-Lorenz relation as it is so far one of the most widely used approaches in the simulation of hybrid and isolated GST nanostructures [40, 56]. Comparing to other EMTs such as the Maxwell-Garnett approximation [60], the maximum difference in the real part of the refractive index Δn_0 between different EMTs was found to be smaller than 5%. Therefore, our estimation is well within the acceptance of EMT approximation.

Technical comment 4) *The authors should discuss more clearly, how they determine the crystallinity C as a function of the annealing time. To me, this aspect is missing and crucial for the*

validity of the presented results. A SEM image as well as an AFM scan of the nanodisks after different annealing times should be added and could support the presented evaluation.

Our response: We thank the reviewer for the valuable comment. We adopted the same treatment as in Ref. 40 to estimate the crystallinity C of the GST disks, that is, we simulate the extinction spectra of GST disks at corresponding crystallinity to match the spectral position of anapole states in the experiment. We note that this evaluation approach and present results are highly robust and reproducible for different samples in our experiment (e.g. Fig. 4b and 4c). Based on this evaluation, we can estimate the relationship between the crystallinity C and the annealing time (see Figure R10), which follows the same trend as in Fig. 3d in Ref. 40.

Figure R10. Relationship between the estimated crystallinity C of GST disks and the annealing time.

Besides, as suggested by the reviewer, we have attempted to investigate the GST nanodisks using SEM and AFM after different annealing time. We aware that thermal annealing of GST would introduce a volume reduction of around 6% between amorphous and crystalline GST films [*J. Appl. Phys.* **86**, 5879, (1999)] and we do see similar changes in our experiment, as depicted in Figure R11. The AFM measurements show that the lateral size and the surface topology of the GST disks do not change while the height is reduced by roughly 5% between the amorphous and crystalline states.

Figure R11. (a, b) AFM images of the same GST nanodisk at the amorphous (a) and crystalline (b) states. (c) The AFM scans at the center of the GST disk. The red and black lines represent the situations for amorphous and crystalline states, respectively.

However, we are not able to quantify this phenomenon as an effective indicator to determine the crystallinity C . This is because it is not clear whether such a change evolves linearly in time. Additionally, an accurate measurement of the small change ($< 6\%$) in height with respect to annealing time is beyond the resolution of our facilities. A thorough investigation of this topic would be an interesting study and we hope our work can stimulate such a development. As a final

remark, we have studied the influence of the height reduction in the simulation and we do not see a substantial impact on the spectral position of the resonances (Figure R12). This is because the diameter-to-height ratio of the GST disks in our study is already high ($D/H \sim 10$) enough, therefore the spectral positions of the anapole and ED states would not change dramatically.

Figure R12. Simulated extinction spectra of a cGST nanodisk with $D = 2 \mu\text{m}$ and different heights H .

In the revised manuscript, we have added the above discussions in the main text and the Supplementary Note 5 as:

Here we estimate the crystallinity C by matching the spectral position of the anapole states in the experiment, similar to Ref. [40].

We also note that the phase change of the GST material may introduce a volume reduction of $\sim 6\%$ from the amorphous to the crystalline state in thin film [S7]. However, it is not clear how such a change would behave and evolve in 3D GST nanostructures, i.e., whether it occurs homogeneously along all the directions and linearly in time or not. In our study, we observed a 5% height reduction between amorphous and crystalline GST nanodisks without any noticeable changes in their lateral sizes or surface topology (Supplementary Fig. 7a). Given the large diameter-to-height ratio of the disks in our study, such a subtle change alone would not influence the spectral position or strengths of the resonances (Supplementary Fig. 7b).

Supplementary Figure 7. (a) The AFM scans at the center of the GST disk at the amorphous (black) and crystalline (red) states. (b) Simulated extinction spectra of a cGST nanodisk with $D = 2 \mu\text{m}$ and different heights H .

Technical comment 5) Overall, the title of the submitted manuscript has to be reviewed since it connotes a different content to the reader meaning that one expects results including reversible switching by the means of optical or electrical pulses.

Our response: We thank the reviewer for the comment. We have deleted the word “dynamic” in the title and revised it as “Active control of anapole states by structuring the phase-change alloy $\text{Ge}_2\text{Sb}_2\text{Te}_5$ ”, which we hope now can reflect the major contents of our work more precisely.

Technical comment 6) The terms “tunability”, “switchability” and “reversibility” should be clearly defined, carefully used and not be mixed up in the manuscript to avoid any misguidance of the reader.

- The variation of crystallinity C can be interpreted as tunability, whereas switchability does not seem to be suited in terms of “mode switching”. In this sense, a control via optical or electrical pulses can be easily associated, which is misleading. “Mode shifting” might be a better suited term.

- The term “reversibility” should be used carefully as well. In the presented work, the crystallinity is changed via annealing, which is to some extent a controllable and reproducible process. This approach does not allow for targeting any desired C , i.e. only an increase of C is possible, which is in contrast to optical or electrical switching. In the given geometry, a reversible targeting of C between 0 and 100% with optical or electrical pulses is not straightforward and should be discussed here. This aspect is missing and thus, the authors promise more functionality than what could be realized with the presented layout. The outlook towards metadevices or reversible cycling of the GST-225 structural phase even, needs to take into account, that many changes on the sample layout would be necessary to realize reversibility. These adaptations might hinder the detection of the anapole modes. The authors need to take this into account to not misguide the reader.

Our response: We thank the reviewer for pointing out the inappropriate use of the terms.

1) We have removed the terms “switchability” or “switching” when describing the shifts of the Mie resonances in the spectra. Instead, we used the term “mode shifting” as suggested by the reviewer. We have marked all the corrections throughout the revised manuscript. Below are some revised descriptions in the abstract and introduction:

By harnessing the dramatic optical contrast of GST, we realize broadband ($\Delta\lambda/\lambda \sim 15\%$) mode shifting from an electric dipole resonance to an anapole state. Active control of higher-order anapoles and multimodal tuning are also investigated...

In this work, we realize broadband and controllable mode shifting between distinct scattering states by structuring the phase change alloy $\text{Ge}_2\text{Sb}_2\text{Te}_5$ (GST).

By exploiting the intermediate phases of GST, we show progressive mode shifting between scattering bright and dark states over an extremely broadband region ($\Delta\lambda/\lambda \sim 15\%$). Multimodal shifting among higher-order ED and anapole states is also manifested...

2) We agree with the reviewer that realizing reversibility in our current layout is not straightforward as most of the studies so far have only reported reversible tuning of GST thin films. Therefore, we have carefully reexamined the use of the terms “reversibility” in our study and deleted the promise of “reconfigurability” in the introduction as:

...our findings establish a new basis for designing active optical components and pave the way towards "metadevices" with tunability and reconfigurability on demand [58].

We have also followed the reviewer's suggestion and added a few lines in the last section to discuss the future adaptations and optimizations that need to be considered for reversible tuning in GST nanostructures:

We note that our present work provides a systematic yet prototypical demonstration of the active multipolar effects in GST nanostructures via thermal annealing. To realize reversible tuning with electric or optical stimuli, further considerations and adaptations of the layout need to be taken into account, for example, the inclusion of electrodes and protective layers [39].

Technical comment 7) *Please consider the following technical comments as suggestions towards a complete technical description and an easily readable publication:*

7a) *What is the size of the probed sample area (number of disks in the array, FTIR microscope aperture size)? Moreover, more FTIR measurement details are missing, which are crucial for reproducing and rating the presented results, such as background normalization, averaged number of scans, resolution and applied objective. Please add these details to the Methods section.*

Our response: We thank the reviewer for the valuable comment. The size of the probed area is $40 \times 40 \mu\text{m}^2$, corresponding to 64 disks in the array with a $3 \mu\text{m}$ gap distance. Other details about the FTIR measurement now have been added to the Methods section as follows:

Transmission spectra were measured by using an infrared microscope (Hyperion1000) coupled to a Fourier transform infrared spectrometer (FTIR, Vertex 70). The detector used in the measurement is an MCT detector integrated into the microscope. The size of the probed area is $40 \times 40 \mu\text{m}^2$, corresponding to ~ 64 disks in the array with a $3 \mu\text{m}$ gap distance. All the measurements represent an average of 16 scans taken at a resolution of 4 cm^{-1} . To determine the transmittance (T) of the GST arrays, air was used as the reference. The experimental extinction spectra were then derived as $1 - T$. A $15\times$ Schwarzschild objective is applied which operates at $\sim 16.7^\circ$ off-normal to the surface of the sample and has a collection cone apex angle of $\pm 7^\circ$.

7b) *The MQ and EQ modes can not (easily) be seen in figures 1 and S2. Either leave them out or mark them in the figures.*

Our response: To provide a clearer presentation, we have removed the EQ mode from the legends and marked the MQ mode in the figures.

7c) *Please rephrase or elaborate on the term "[...] with any crystalline phases of GST." used on page 6.*

Our response: We have rephrased the term in the revised manuscript as:

Interestingly, such an amount is nearly constant over the whole spectral range, meaning that the presented mode shifting functionality (from ED to anapole states) can be attained by simply introducing a fixed phase change ($\Delta C \sim 25\%$) to the GST nanosphere with an arbitrary crystallinity C below 75%.

7d) *Please comment on the term "dispersionless tunability" on the top of page 8.*

Our response: We have rephrased the term as “(almost) nondispersive behavior” and elaborated on it as follows:

Therefore, given a cluster of GST spheres with various radii, by introducing a fixed amount of phase change ($\Delta C = 25\%$), all the structures possessing different ED resonance wavelengths would exhibit the *same* functionality shifting from ED to corresponding anapole states. Thus, the mode shifting effects in GST nanostructures can be maintained in a broadband range regardless of the original resonance wavelengths. Such an almost “non-dispersive” behavior may find its applications in many interesting aspects. For instance, a major challenge nowadays to realizing actively tunable metasurfaces lies in the fact that metasurfaces are usually composed of plasmonic or dispersive meta-atoms with different sizes and different resonant wavelengths. Therefore, a uniform optical change across the interface does not guarantee that the metasurface can sustain its important functions (e.g., focusing, invisibility, polarization conversion, etc) after active tuning. By contrast, GST resonators with nearly non-dispersive behaviors may provide a promising solution to overcome this issue.

7e) Please add a reference to the refractive index of CaF2 on page 13.

Our response: We have added the following paper in the reference and cite it in the Methods section as:

The refractive index of the substrate CaF2 was set to 1.4 [62].

[62] Li. H. Refractive index of alkaline earth halides and its wavelength and temperature derivatives. *J. Phys. Chem. Ref. Data* **9**, 161–290 (1980).

7f) Please correct the caption of figure S3: “[...] We mention that the Fig. S3c is the same [...].”

Our response: We thank the reviewer for pointing out the inconsistency and we have now corrected the caption of figure S3 (now Supplementary Fig. 5).

7g) Typo on page 14: “Vertex70”

Typo on page 15: “Cambridge University Press”

Our response: We thank the reviewer for pointing out the typos. Now they have been corrected in the manuscript.

General Comment 2) In conclusion, I recommend a resubmission after major revisions.

Our response: We once again thank the reviewer for his/her insightful comments on our work.

Reviewers' Comments:

Reviewer #2:

Remarks to the Author:

In this revised version, the authors have answered all comments raised by the referees in a satisfactory manner. I only found a few typos as listed below. I can thus now recommend the manuscript for publication in Nature Communications.

minor typos:

page 2, middle: "asscoiated maximum in the near-field energy" -> "associated"

page 6, middle: "white dahsed lines" -> "dashed"

page 15, sample characterization: "used as the referenc" -> "reference"

page 15, sample characterization: "implemeneted progressively" -> "implemented"

Reviewer #3:

Remarks to the Author:

Thank you very much for the efforts responding to my comments. The changes and the added material improved the manuscript a lot. I recommend a publication of this work.

Reviewer #2

In this revised version, the authors have answered all comments raised by the referees in a satisfactory manner. I only found a few typos as listed below. I can thus now recommend the manuscript for publication in Nature Communications.

minor typos:

page 2, middle: "asscoiated maximum in the near-field energy" -> "associated"

page 6, middle: "white dahsed lines" -> "dashed"

page 15, sample characterization: "used as the referenc" -> "reference"

page 15, sample characterization: "implemeneted progressively" -> "implemented"

Our response: We thank the reviewer for carefully reading our work. We have read through our manuscript and made corresponding corrections to ensure clarity in language and grammar.

Reviewer #3

Thank you very much for the efforts responding to my comments. The changes and the added material improved the manuscript a lot. I recommend a publication of this work.

Our response: We thank the reviewer for his/her recommendation of our work.